# Visible light-triggered selective C(sp²)-H/C(sp³)-H coupling of benzenes with aliphatic hydrocarbons

Qian-Yu Li[1], Shiyan Cheng[1], Ziqi Ye[1], Tao Huang[1], Fuxing Yang[1], Yu-Mei Lin ◎[1] ✉ & Lei Gong ◎[1,2] ✉

The direct and selective coupling of benzenes with aliphatic hydrocarbons is a promising strategy for C(sp²)-C(sp³) bond formation using readily available starting materials, yet it remains a significant challenge. In this study, we have developed a simplified photochemical system that incorporates catalytic amounts of iron(III) halides as multifunctional reagents and air as a green oxidant to address this synthetic problem. Under mild conditions, the reaction between a strong C(sp²)-H bond and a robust C(sp³)-H bond has been achieved, affording a broad range of cross-coupling products with high yields and commendable chemo-, site-selectivity. The iron halide acts as a multifunctional reagent that responds to visible light, initiates *C*-centered radicals, induces single-electron oxidation to carbocations, and participates in a subsequent Friedel-Crafts-type process. The gradual release of radical species and carbocation intermediates appears to be critical for achieving desirable reactivity and selectivity. This eco-friendly, cost-efficient approach offers access to various building blocks from abundant hydrocarbon feedstocks, and demonstrates the potential of iron halides in sustainable synthesis.

The direct and selective coupling of benzenes and low reactive aliphatic hydrocarbons, including alkanes and cycloalkanes, has garnered significant attention in the field of organic synthesis as an ideal strategy for C(sp²)-C(sp³) bond formation[1–5]. This method offers advantages in terms of both atoms and steps, while also benefiting from the abundance of starting materials. However, achieving desirable reactivity, chemo- and site-selectivity during the C-H cleavage and subsequent C-C formation is a challenging task due to the intrinsic inertness of various C-H bonds in these hydrocarbon feedstocks, as well as their high similarity in bond strength and chemical environment. For instance, benzene possesses a bond dissociation energy (BDE) of 110 kcal/mol, while (cyclo)alkanes typically have BDEs exceeding 96 kcal/mol[6,7]. Reactions of substituted benzenes involve *ortho-*, *meta-* and *para-*selectivity, whereas branched (cyclo)alkanes contain various primary, secondary, and tertiary C-H reaction sites (Fig. 1a)[8–11].

Side reactions such as polyalkylations, rearrangements, and self-couplings are also critical issues that must be taken into account[12–15]. Consequently, developing efficient systems to facilitate selective coupling reactions of unactivated C-H bonds, beyond those relying on directing groups or Minisci-type alkylation of heteroarenes[16–22], represents an insurmountable challenge requiring a deep understanding of the complex interplay between reactivity, selectivity, and reaction conditions.

Despite significant efforts, very few strategies relying on functional catalysts, such as metal-modified zeolite, montmorillonite and heteropoly acid catalysts[23,24], have shown potential in this aspect (Fig. 1b). For example, Goldman et al. demonstrated that the combination of pincer-ligated iridium catalysts and zeolites enabled an intramolecular reaction at 205 °C of an alkyl-H and an aryl-H bond in *n*-pentylbenzene, yielding 1-methyl-1,2,3,4-tetrahydronaphthalene

¹Key Laboratory of Chemical Biology of Fujian Province, College of Chemistry and Chemical Engineering, Xiamen University, Xiamen, Fujian 361005, China. ²Innovation Laboratory for Sciences and Technologies of Energy Materials of Fujian Province (IKKEM), Xiamen 361005, China. ✉e-mail: linyum@xmu.edu.cn; gongl@xmu.edu.cn

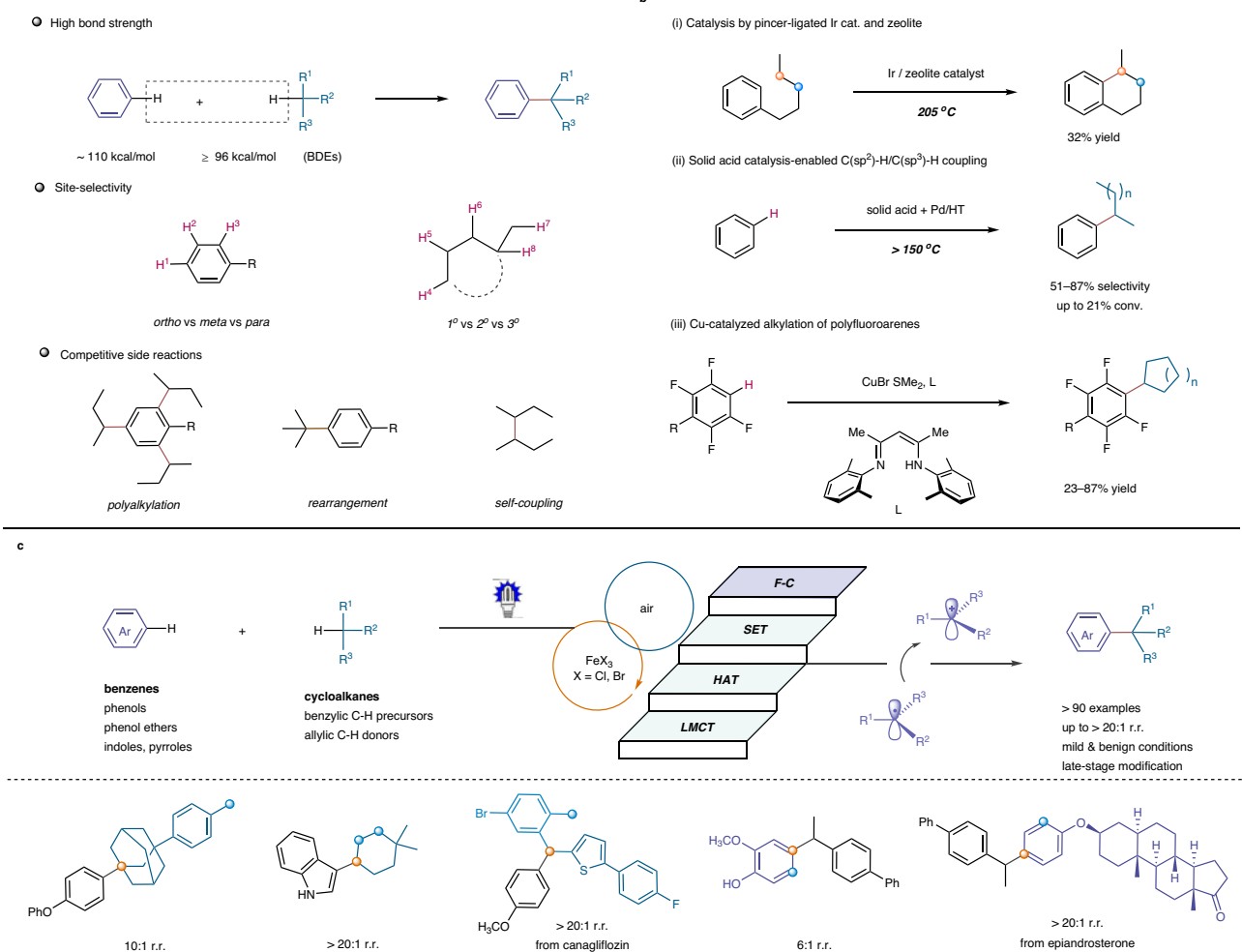

**Fig. 1 | Overview of this work. a** Challenges associated with cross-coupling between a C(sp²)-H bond in arenes and a C(sp³)-H bond in alkanes. **b** Representative studies on cross-coupling between a C(sp²)-H bond in arenes and a C(sp³)-H bond in alkanes. **c** This work: iron(III) halide-enabled photochemical selective coupling of benzene derivatives and C(sp³)-H hydrocarbons. BDE bond dissociation energy, LMCT ligand-to-metal charge transfer, HAT hydrogen atom transfer, SET single electron transfer, F-C Friedel-Crafts-type process, r.r. regioisomeric ratio.

in up to 32% yield[25]. Motokura et al. reported a direct C-H alkylation of benzene with alkanes and cycloalkanes at 150 °C for 64 h in the presence of montmorillonites as solid acid catalysts, which provided with 6–22% conversions and regioselectivities of 51–87%[26,27]. Chang et al. developed a copper-catalyzed C-H alkylation of polyfluoroarenes, a class of specific arenes of extremely electron-deficient features, with (cyclo)alkanes or other alkylating sources, delivering alkylbenzene products in 23–87% yields and with moderate to good regioselectivity[28]. New approaches to promote such transformations in a more sustainable and general way are strongly demanded.

Iron salts and complexes have long been recognized as attractive reagents and catalysts in organic synthesis, owing to the abundance of iron element on earth and its biocompatibility[29–31]. Recent studies have revealed the potential of iron salts in photochemical reactions, with the ability to activate and break aliphatic C-H bonds[32–36]. Building upon our experience with visible light-driven photochemical synthesis[37–39], we aimed to develop an economical and environmentally friendly approach for the direct coupling of low-reactive C(sp²)-H and C(sp³)-H bonds under mild conditions (Fig. 1c). Our mechanistic hyphosis involves the use of a catalytic amount of an iron(III) halide (X = Cl, Br) to initiate the formation of alkyl radicals derived from C(sp³)-H precursors. The resulting radicals are then oxidized by Fe(III), generating carbocation intermediates that undergo the subsequent Friedel-Crafts-type process to furnish alkylarene products. Air, acting as a

mild oxidant, can oxidize the reduced iron species and regenerate Fe(III) for the next cycles. The gradual release of radical species and carbocations, along with the low concentrations of active intermediates, allow for useful reactivity and selectivity. Here, we show an iron(III) halide-enabled photochemical aerobic dehydrogenative coupling reaction between C(sp²)-H precursors (e.g., benzenes, phenols, phenol ethers, indoles, and pyrroles) and C(sp³)-H donors (e.g., benzylic, allylic C(sp³)-H derivatives, and cycloalkanes). This method provides straightforward access to valuable alkylarenes with high chemo- and site-selectivity under extremely mild reaction conditions, and offers an appealing strategy for constructing complex organic molecules using low-cost starting materials and environmentally benign iron salts.

## Results and discussion
### Initial experiments
With the aim of investigating iron-promoted photochemical cross-coupling between low reactive C(sp²)-H and C(sp³)-H bonds, we selected anisole (**1a**) and 4-ethylbiphenyl (**2a**) as model substrates and utilized air as an environmentally friendly oxidizing agent. Various metal salts, including CoCl₂, NiCl₂, CuCl₂, Fe₂(SO₄)₃, Fe(NO₃)₃, FeCl₃ or FeBr₃, were tested as initiators in catalytic amounts. The results indicated that only iron halides displayed activity in this transformation (Fig. 2, entries 1–9). Specifically, when **1a** and **2a** were reacted in 1,1,2,2-tetrachloroethane (TCE) with 20 mol% FeCl₃·6H₂O under an air

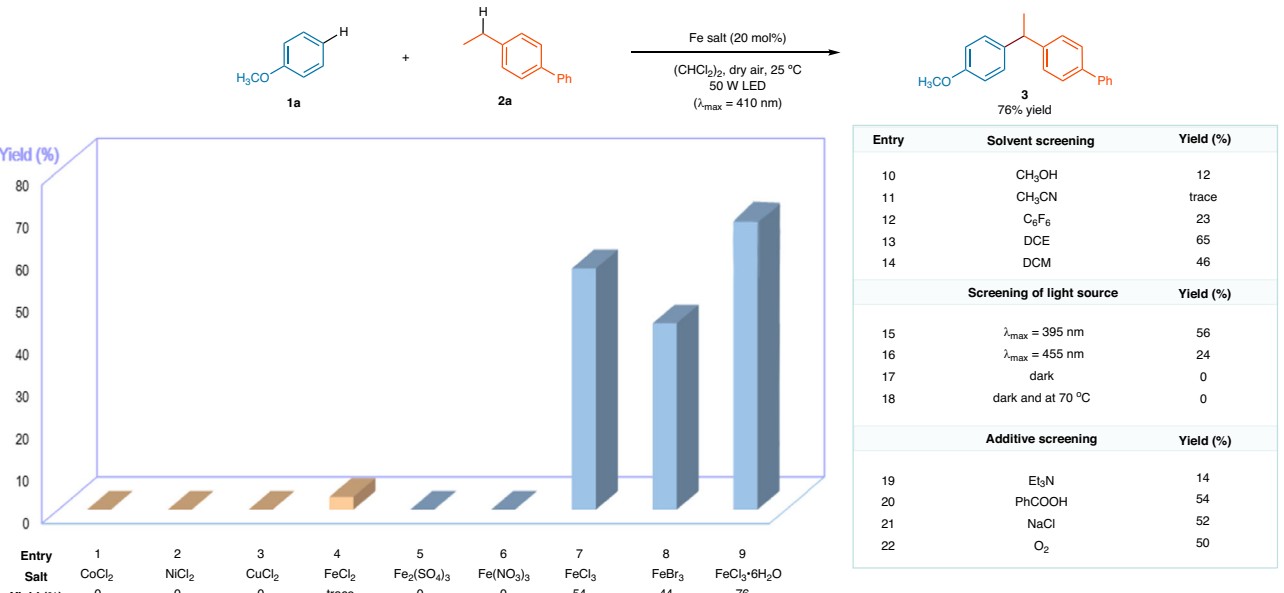

**Fig. 2 | Optimization of reaction conditions.** Standard conditions: **1a** (0.30 mmol), **2a** (0.90 mmol), Fe salt (0.060 mmol), indicated solvent (0.75 mL), dry air, irradiation with a 50 W LED lamp ($\lambda_{max}$ = 410 nm), 25 °C, 48 h.

atmosphere and irradiation with a 50 W LED lamp ($\lambda_{max}$ = 410 nm), the alkylbenzene product (**3**) was obtained in a good yield of 76% (entry 9). Solvent screening experiments were performed, and it was found that the yields were found to decrease when using other solvents such as $CH_3OH$, $CH_3CN$, $C_6F_6$, dichloroethane (DCE), or dichloromethane (DCM) (entries 10–14). Different light sources, such as a 50 W LED lamp with maximum wavelength at 395 or 455 nm, provided yields of 56% and 24%, respectively (entries 15, 16). Moreover, the reaction did not proceed in the dark at either 25 °C or 70 °C (entries 17, 18), highlighting the critical role of light irradiation with a suitable wavelength for the transformation to occur. External additives, including an amine base, a carboxylic acid, a chloride source and oxygen were also examined, but none of them enhanced the yield (entries 19–22).

## Substrate scope

Under optimal conditions, we conducted an assessment of the substrate scope of C(sp²)-H precursors in the iron halide-enabled photochemical reaction (Fig. 3). Our study included a range of phenyl ethers (products **3–6**), substituted phenyl ethers (**7–11**), 2,3-dihydrobenzofuran (**12**), benzo[*d*][1,3]dioxole (**13**), 2,3-dihydrobenzo[*b*][1,4]dioxine (**14**), phenyl sulfides (**15–17**), phenol and its derivatives (**18–25**), all of which were found to be compatible. The corresponding products were obtained with yields ranging from 43% to 76%, typically with only a single regioisomer observed. This process demonstrated high regioselectivity towards competitive C(sp²)-H sites. For example, during the formation of compound **25**, the 4- and 5-position of Guaiacol were selectively benzylated in a satisfactory ratio of 6.0:1. Additionally, product **26**, bearing three reactive sites in the substrate, was formed in a 66% yield and displayed exclusive regioselectivity. It is worth noting that benzene was successfully converted into the desired product (**27**) in a 45% yield. Alkyl benzenes containing competitive benzyl C-H bonds acted as C(sp²)-H precursors instead of C(sp³)-H donors in this reaction. The reaction of toluene or *tert*-butyl benzene with 1-bromo-4-ethylbenzene gave the desired products (**28, 29**) with *para:ortho* ratios of 3.0:1 and 4.0:1, respectively, while those of biphenyls yielded the corresponding products (**30–32**) as single regioisomers in 45–53% yield.

Aromatic *N*-heterocycles are widely present in bioactive molecules and natural products, with their benzylated derivatives attracting significant attention due to the potential therapeutic applications[40]. Under

standard conditions, reactions of 1*H*-indole, 7-bromo-1*H*-indoles or 5-bromo-1-methyl-1*H*-indole resulted in the formation of 3-benzylated indoles (**33–36**) with yields ranging from 49% to 61%. *N*-acyl protected indoles, such as 1-(1*H*-indol-1-yl)ethanone (product **37**) and its derivatives (**38–42**), were found to give typically higher reaction yields (64–73%). The reaction of an *N*-tosyl protected pyrrole selectively occurred at 2-position, furnishing the product (**43**) in a 45% yield. These findings indicate a high tolerance of this method towards indole- and pyrrole-based heterocycles.

In terms of C(sp³)-H precursors, we first explored various electronically and sterically diverse ethylbenzene derivatives (Fig. 4). These reactions, under standard conditions, produced coupling products (**44–57**) with yields of 55–74%. Additionally, alkyl benzenes with a long side chain (**58, 59**), fused aliphatic rings (**60–62**), fused aromatic rings (**63**), as well as tertiary or primary benzylic C-H bonds (**64–67**), were found to be compatible, delivering products with yields of 49–65%. To demonstrate the precise C-H recognition of this method, several substrates containing competitive primary, secondary, and tertiary benzylic C-H bonds were tested. For instance, 1-(*p*-tolyl)adamantane, a compound with both benzyl and adamantanyl C-H bonds, was selectively arylated at the adamantanyl C-H site of higher bond strength, affording the corresponding product (**68**) in a 52% yield and with 10:1 r.r.. The Hirshfeld charge analysis revealed that the carbocation on the alkyl group had more positive charges, which could be advantageous for further alkylation steps (see more details in Supplementary Information Section 6.2). The reaction with 1-ethyl-4-methylbenzene exhibited a site-selectivity of 4.8:1 r.r. (product **69**), whereas 1-ethyl-4-(4-isopropylphenoxy)benzene or 1-cyclohexyl-4-ethylbenzene provided with 3.0:1 and 10:1 r.r. (products **70, 71**), respectively. In these instances, the C(sp³)-H arylation was observed to predominantly take place at secondary C-H bonds, rather than primary or tertiary ones. This selectivity can be attributed to a delicate balance between steric hindrance and electronic preferences[41], as well as the influence of iron salts in other steps of the process.

Allylic C-H donors were identified as suitable substrates and could be selectively coupled with benzenes. For example, (2-methylprop-1-en-1-yl)benzene, which bears two closely related allylic C-H bonds, exhibited a preference for undergoing C-H arylation of the (*Z*)- over (*E*)-methyl group. This resulted in the formation of the product (**74**)

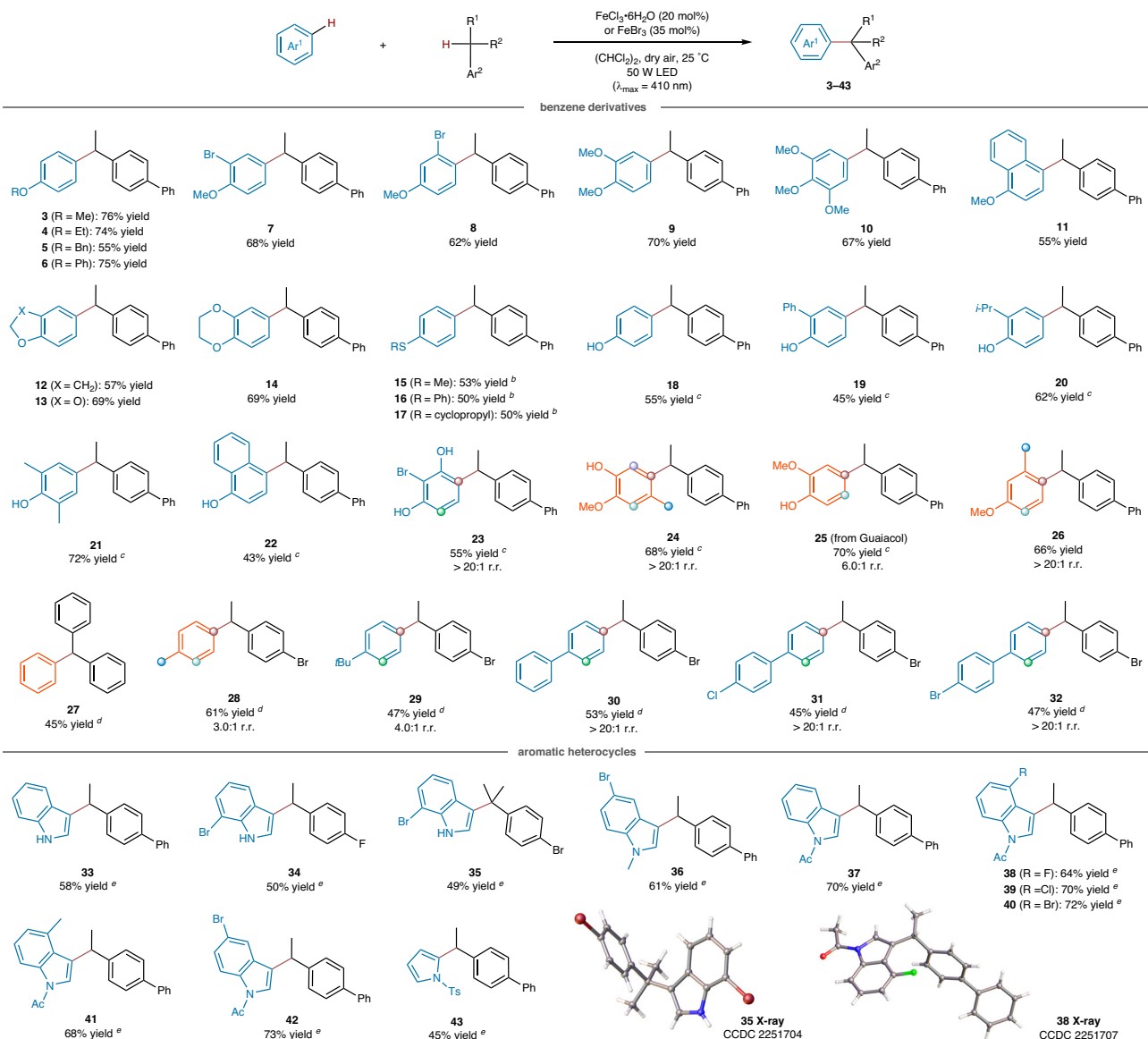

**Fig. 3 | Substrate scope of C(sp²)-H precursors.** Standard conditions: arenes (0.30 mmol), C(sp³)-H donors (0.90 mmol), FeCl₃ · 6H₂O (0.060 mmol), (CHCl₂)₂ (0.75 mL), dry air, irradiation with a 50 W LED lamp (λ_max = 410 nm), 25 °C, 48 h. Isolated product yields. [a] Reaction in (CH₂Cl)₂. [b] Reaction in CHCl₃. [c] Reaction in the presence of FeBr₃ (0.11 mmol), (CH₂Cl)₂ (0.75 mL). [d] Reaction in CH₂Cl₂.

with a yield of 58% and > 20:1 r.r.. Despite having higher bond dissociation energies of C-H bonds compared to toluene derivatives and allylic C-H precursors, various cycloalkanes and their substituted derivatives were shown to be applicable in the reaction. Adamantane, 1-methyl adamantine, cyclopentane, cyclohexane, cycloheptane and 1,1-dimethyl cyclohexane, were successfully converted into the coupling product (**75–80**) in 40–49% yield under slightly modified conditions. Notably, adamantane and its derivatives have stronger tertiary C-H bonds than secondary C-H bonds (BDE of ~99 kcal/mol vs ~96 kcal/mol), but their reactions exhibited exclusive selectivity towards tertiary C-H bonds, yielding products **75** and **76**.

## Mechanistic studies

To investigate the reaction mechanism, several control experiments were designed and conducted (Fig. 5). The addition of 2,2,6,6-tetramethylpiperidine-1-oxyl (TEMPO) or 2,6-di-*tert*-butyl-4-methylphenol (BHT) to the photochemical reaction of **1d** and **2a** resulted in complete inhibition the formation of product **6** (Fig. 5a, left). Instead, HRMS analysis detected a TEMPO- and a BHT-benzyl radical coupling product

(**81, 82**). When the reaction was carried out in darkness in the presence of TEMPO, compound **81** was not observed. These findings strongly support the hypothesis of a pathway that proceeds through *C*-centered radicals. The participation of chlorine radicals was confirmed by the reaction of *N*-tosyl diallylamine (**83**) and stoichiometric FeCl₃ · 6H₂O under standard conditions, which gave a cyclization product (**84**) with a yield of 61% (Fig. 5a, right)[42]. Moreover, the reaction of **1d** and 4-(cyclopro-pylmethyl)−1,1′-biphenyl (**85**) under standard conditions yielded a ring-opened alkyl chloride (**86**) with a yield of 66%. The reaction between **1d** and cyclopropylbenzene (**87**) under standard conditions with the addition of nucleophilic sodium acetate also afforded a ring-opening product (**88**) in a 77% yield (For an in-depth understanding of the formation mechanism of product **88**, please refer to page S79 of the Supplementary Information). These results suggest that the free radical chain reaction occurs between the chlorine radical and aryl cyclopropane, and the cation is subsequently captured by the nucleophilic reagent, further demonstrating the involvement of chlorine radicals in the photochemical coupling reaction (Fig. 5a, right)[43,44].

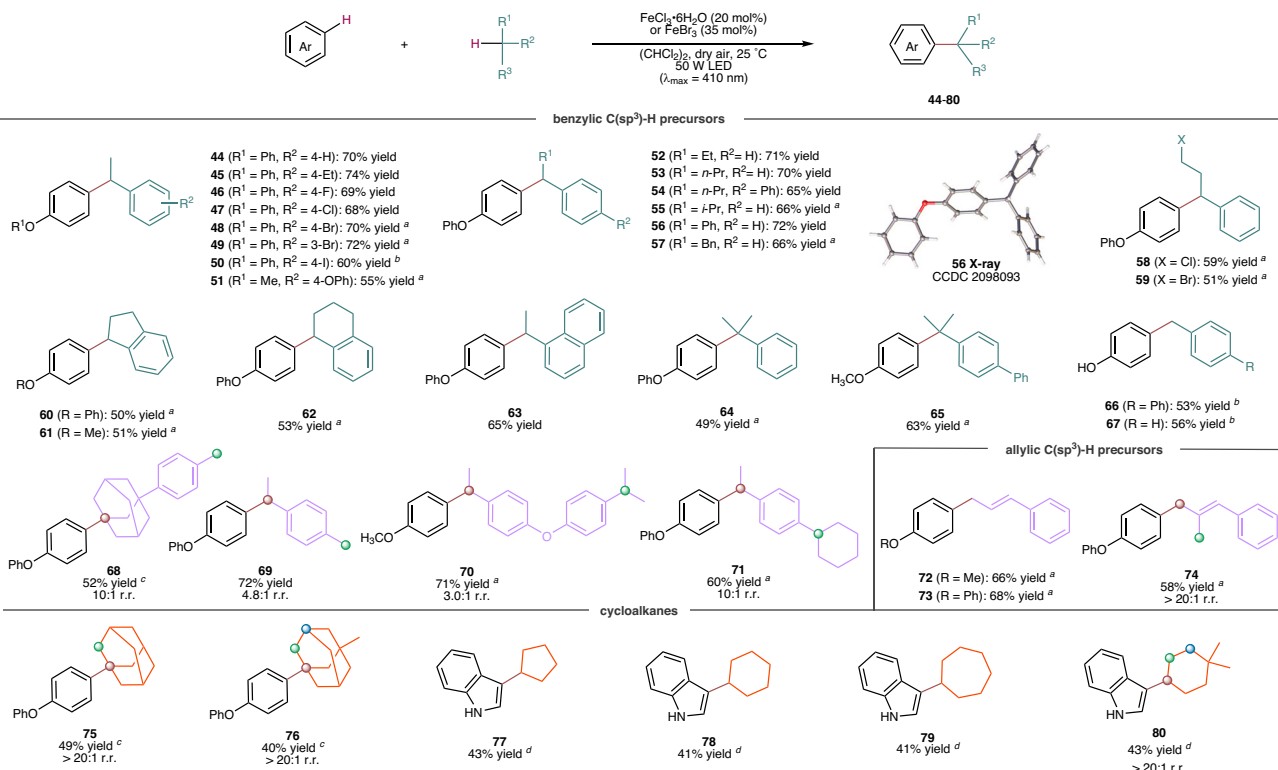

**Fig. 4 | Substrate scope of C(sp³)-H precursors.** Standard conditions: arenes (0.30 mmol), C(sp³)-H precursors (0.90 mmol), FeCl₃ · 6H₂O (0.060 mmol), (CHCl₂)₂ (0.75 mL), dry air, irradiation with a 50 W LED lamp (λ_max = 410 nm), 25 °C, 48 h. Isolated product yields. *a* Reaction in (CH₂Cl)₂. *b* Reaction in CHCl₃. *c* Reaction in the presence of adamantane and its derivatives (3.0 mmol), FeBr₃ (0.11 mmol), (CH₂Cl)₂ (0.75 mL), irradiation with a 40 W LED lamp (λ_max = 370 nm) at 80 °C for 60 h. *d* Reaction in the presence of FeBr₃ (0.11 mmol), cycloalkane (0.75 mL) irradiation with a 40 W LED lamp (λ_max = 370 nm) at 25 °C for 120 h.

The reaction of **1d** + **2a** → **6** exhibited a light-dark interval response, indicating the essential need for continuous light irradiation was essential for the transformation to occur (Fig. 5b). X-ray photoelectron spectroscopy (XPS) analysis was performed on the residue obtained from the reaction of **1d** and **2a**, in the presence of 2.0 equivalent of FeCl₃ · 6H₂O under argon. An evident signal at 709.8 eV, corresponding to Fe²⁺ species, was detected. In contrast, the residue obtained from the reaction under standard conditions showed a significant signal at 711.0 eV, assigned to Fe³⁺ species. These observations imply a possible pathway involving Fe(III)→Fe(II)→Fe(III) and air oxidation, as depicted in Fig. 5c[45]. UV-Vis absorption spectra of the individual components and their various combinations revealed that the substrates (**1d, 2a**), the product (**6**), and different mixtures ([**1d** + **1a**], [**1d** + **1a** + **6**]) displayed absorption only at λ < 330 nm. Meanwhile, the iron salt exhibited evident absorption at λ > 400 nm. These results suggest that the species responsive to visible light could be derived from iron species present within the system (Fig. 5d and Supplementary Fig. 3).

Incorporating nucleophilic sodium acetate or sodium thiocyanate into this photochemical reaction resulted in 1-([1,1'-biphenyl]−4-yl) ethyl acetate (**89**) or 4-(1-thiocyanatoethyl)−1,1'-biphenyl (**90**) in yields of 47% and 69%, respectively, indicating a potential pathway via carbon cations (Fig. 5e). Irradiating 4-ethyl-1,1'-biphenyl (**2a**) with 2.0 equivalent of FeCl₃ • 6H₂O under argon yielded a chlorinated product (**91**) in a yield of 44% and a self-coupling product (**92**) in a yield of 49%. It is worth mentioning that these products did not form in the dark. In the presence of **1d**, a C(sp²)-H precursor, (1-chloroethyl)benzene or its bromide analog could be easily converted to the cross-coupling product (**44**) in a good yield (Fig. 5f). These findings suggest that the reaction can proceed through C-H chlorinated intermediates, but light irradiation is essential for their generation. Furthermore, kinetic isotope effect (KIE) experiments estimated the $K_H/K_D$ ratio of the reaction

**1d** + **2b** → **44** as 2.62, confirming that the hydrogen atom abstraction is the rate-determining step (Fig. 5g).

Based on these mechanistic experiments and reported literatures[34,36,46], we proposed a plausible mechanism (Fig. 5h). Initially, visible light induces a ligand-to-metal charge transfer (LMCT) of the iron salt (FeCl₃·6H₂O or FeBr₃), generating FeCl₂ and a chlorine radical. This active radical abstracts a hydrogen atom from an aliphatic C-H bond, affording HCl and a *C*-centered radical. Aerobic oxidation of the iron(II) species regenerates the iron(III) salt, which can further oxidize the *C*-centered radical to a carbon cation[47-49]. Subsequently, the C(sp²)-C(sp³) cross-coupling coupling of the alkyl carbon cation, or a corresponding chloride derivative from nucleophile attack, with arenes leads to product formation. The final step can be viewed as a Friedel-Crafts-type process promoted by an iron-based Lewis acid.

## Synthetic applications

With a more comprehensive understanding of the reaction mechanism, we conducted an evaluation of the synthetic utility of this method (Fig. 6). Our investigation revealed that a diverse array of derivatives from natural products and medicinal relevance were compatible with this protocol. For instance, 2-(phenoxymethyl)tetrahydrofuran (product **93**), 1-(phenoxymethyl)adamantane (**94**), sesamol (**95**), a *L*-menthol derivative (**96**), a (+)-fenchol derivative (**97**), a *L*(-)-borneol derivative (**98**) and a derivative of epiandrosterone (**99**) all effectively react with **2a** to yield coupling products (**93–99**) as single regioisomers in yields ranging from 35% to 73%. Additionally, methoxybenzene (**1a**) smoothly react with a derivatized difluoro-benzodioxole, a canagliflozin derivative, or triclosan under standard conditions, leading to the formation of products (**100–102**) in yields of 40–70%. These results further illustrate the high functionality compatibility and selectivity of the reaction.

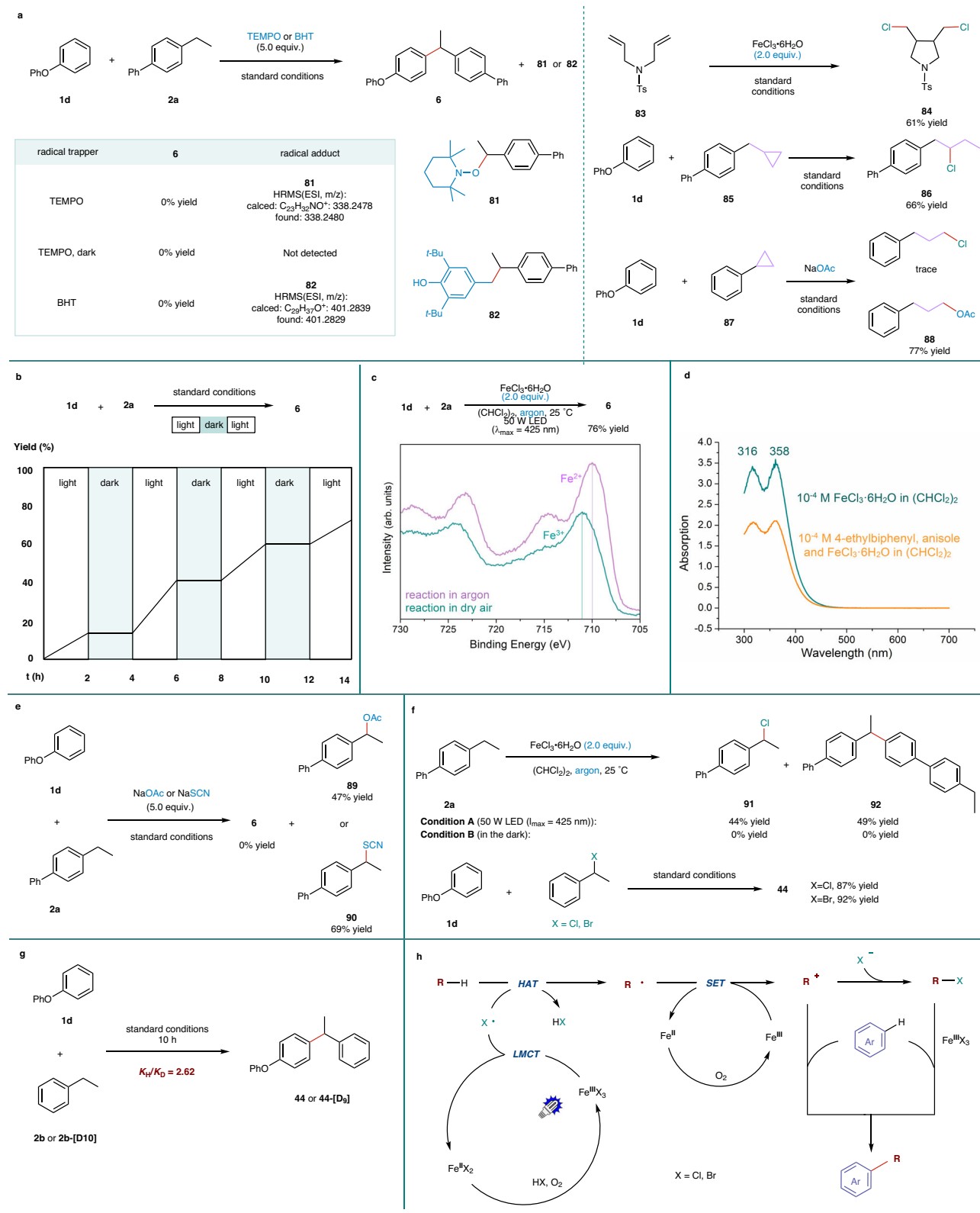

**Fig. 5 | Mechanistic studies. a** Radical trapping experiment. **b** Light on/off experiments. **c** X-ray photoelectron spectroscopy (XPS) analysis. **d** UV-Vis spectroscopy. **e** Reactions interfered with competitive nucleophiles. **f** Probing possible intermediates. **g** Kinetic isotope effect (KIE) experiments. **h** A plausible reaction mechanism.

Finally, several bioactive molecules were prepared on the basis of the photochemical cross-coupling reaction. For instance, Nafenopin is a potent hypolipidemic drug (\$ 435.5/100 mg, Alichem Inc.)[50], and its ester derivative (**105**) can be easily synthesized in just two steps. The

reaction of phenol (\$ 121/2.0 kg, Alfa Aesar) and 1,2,3,4-tetra-hydronaphthalene (\$ 41.9/1.0 kg, Alfa Aesar) was conducted under standard conditions, furnishing 4-(1,2,3,4-tetrahydronaphthalen-1-yl) phenol (**103**) in a 69% yield. Compound **103** was then treated with ethyl

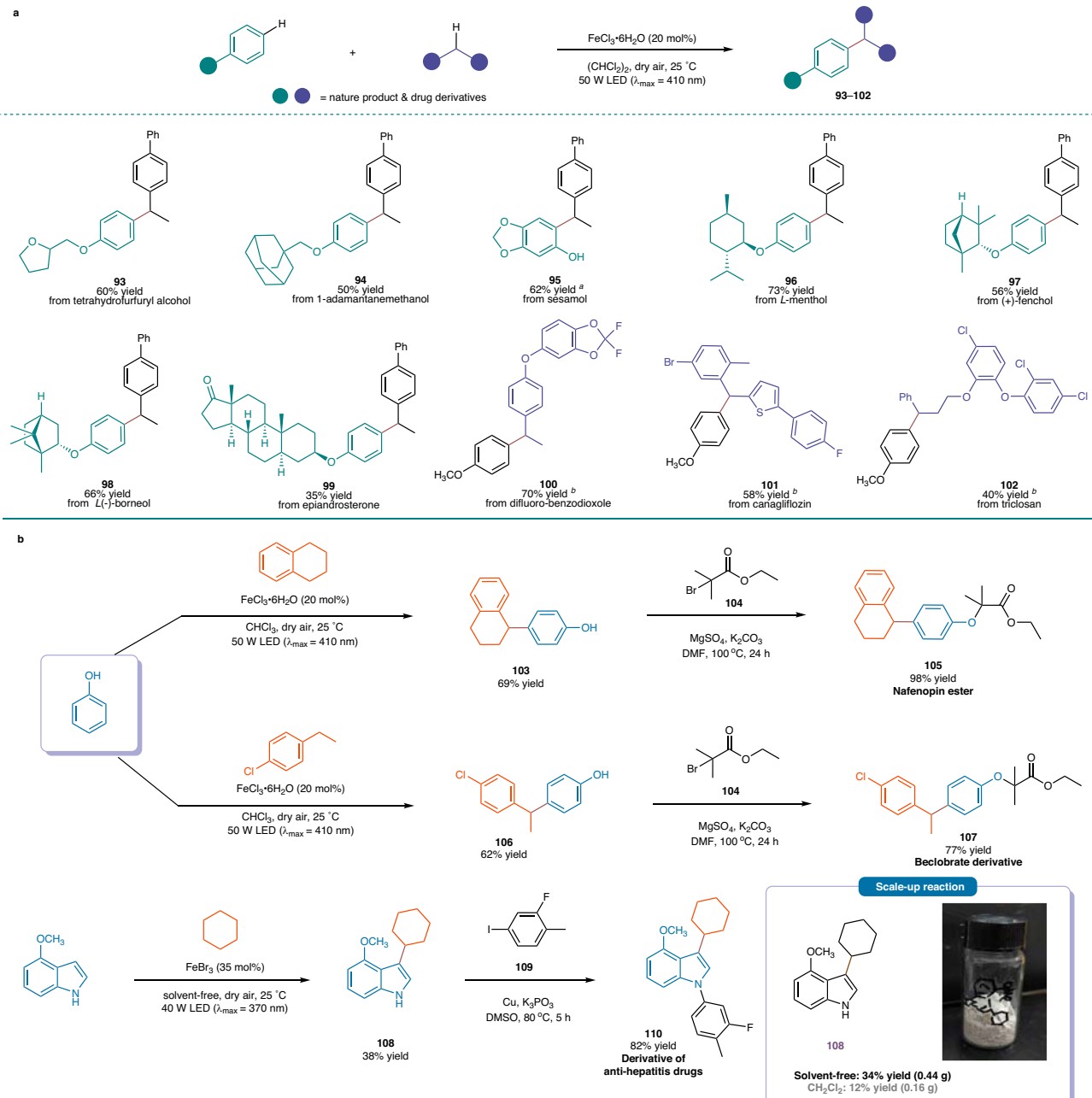

**Fig. 6 | Synthetic applications of this method. a** Late-stage modification of bioactive molecules. Standard conditions: **1zo–1zu** (0.30 mmol), **2a, 2zj–2zl** (0.90 mmol), FeCl$_3$ • 6H$_2$O (0.060 mmol), (CHCl$_2$)$_2$ (0.75 mL), dry air, irradiation with a 50 W LED lamp ($\lambda_{max}$ = 410 nm), 25 °C, 48 h. [a] Reaction in CHCl$_3$. [b] Reaction in (CH$_2$Cl)$_2$. **b** Synthesis of derivatives of biomolecules and a scale-up reaction.

2-bromo-2-methylpropanoate (**104**, $ 69.6/250 g, Alfa Aesar) for 24 h in the presence of K$_2$CO$_3$, and gave the nafenopin ester (**105**) in a 98% yield. This cost-efficient two-step procedure can also be utilized to synthesize a beclorbrate derivative (**107**). When cyclohexane was subjected to the same conditions and reacted with 4-methoxyindole, it resulted in the arylated product (**108**) in a 38% yield. This compound could be subsequently converted into the derivative of an anti-hepatitis drug (**110**) in an 82% yield. Notably, the reaction of 4-methoxyindole (5.5 mmol) with cyclohexane can be easily scaled up, providing a comparable isolated yield (0.44 g, 34% yield).

We have developed a simplified strategy that utilizes multiple functionalities of iron(III) halides under visible light conditions for the direct and selective coupling of low-reactive C(sp$^2$)-H and C(sp$^3$)-H bonds. In the absence of any ligand and with air serving as the green

oxidant, a catalytic amount of FeCl$_3$ or FeBr$_3$ facilitates the reaction of a wide variety of C(sp$^2$)-H precursors including benzenes, phenols, phenol ethers, phenyl sulfides, indoles and pyrroles with C(sp$^3$)-H donors such as benzylic, allylic derivatives and cycloalkanes. This protocol produces various C(sp$^2$)-C(sp$^3$) coupling products in good yields and with high chemo- and site-selectivity. Mechanistic studies reveal that the iron salt is not only involved in the light-triggered initiation of *C*-centered radicals but also in the induction of single-electron oxidation under aerobic conditions, and participates in the subsequent carbocation-mediated alkylation of arenes. The step-by-step release of radical species and carbocation intermediates appears to be the key to success in achieving desirable reactivity and selectivity. Overall, this study provides a promising approach to the selective activation of strong C-H bonds, enabling the construction of valuable C(sp$^2$)-C(sp$^3$) bonds from abundant reserves

of hydrocarbon feedstocks, and will contribute to the development of efficient and sustainable synthetic methods in organic chemistry.

## Methods

### General procedure for the photochemical cross-coupling of arenes with benzylic or allylic C(sp³)-H precursors

A Schlenk tube (10 mL) was charged with arenes (**1a–1zn**, 0.30 mmol) and benzylic or allylic C-H precursors (**2a–2x, 2z–2zd**, 0.90 mmol), FeCl$_3$·6H$_2$O (0.060 mmol) or FeBr$_3$ (0.11 mmol), (CHCl$_2$)$_2$ (or (CH$_2$Cl)$_2$, CHCl$_3$, CHCl$_2$, 0.75 mL). The mixture was degassed via three freeze-pump-thaw cycles, then filled with dry air. The Schlenk tube was positioned approximately 5 cm away from a 50 W LED lamp ($\lambda_{max}$ = 410 nm). After being stirred at 25 °C for 48 h, the reaction mixture was concentrated to dryness. Purification using silica gel column chromatography gave the pure products.

### General procedure for the photochemical cross-coupling of arenes with cycloalkanes

A Schlenk tube (10 mL) was charged with arenes (**1d, 1ze** or **1zv**, 0.30 mmol) and cycloalkanes (**2 y, 2ze–2zj**, 3.0 mmol or 0.75 mL), FeBr$_3$ (0.11 mmol) and (CH$_2$Cl)$_2$ (0.75 mL) or free-solvent. The mixture was degassed via three freeze-pump-thaw cycles, then filled with dry air. The Schlenk tube was positioned approximately 5 cm away from a 40 W LED lamp ($\lambda_{max}$ = 370 nm). After being stirred at 80 °C for 60 h or at 25 °C for 120 h, the reaction mixture was concentrated to dryness. Purification using silica gel column chromatography gave the pure products.

### General procedure for the late-stage modification of nature product derivatives and drug-like molecules

A Schlenk tube (10 mL) was charged with arenes (**1a, 1zo–1zu**, 0.30 mmol), benzylic C-H precursors (**2a, 2zk–2zm**, 0.90 mmol), FeCl$_3$·6H$_2$O (0.060 mmol), ((CHCl$_2$)$_2$ or (CH$_2$Cl)$_2$, 0.75 mL). The mixture was degassed via three freeze-pump-thaw cycles, then filled with dry air. The Schlenk tube was positioned approximately 5 cm away from a 50 W LED lamp ($\lambda_{max}$ = 410 nm) or a 40 W LED lamp ($\lambda_{max}$ = 370 nm). After being stirred at 25 °C for 48 h, the reaction mixture was concentrated to dryness. Purification using silica gel column chromatography gave the pure products.

## Data availability

The authors declare that the data supporting the findings of this study are available within this article and its Supplementary Information file, or from the corresponding authors upon request. The experimental procedures and characterization of all new compounds are provided in Supplementary Information, and coordinates of the optimized structures are provided as source data. The X-ray crystallographic coordinates for structures reported in this study have been deposited at the Cambridge Crystallographic Data Center (CCDC), under deposition numbers CCDC 2251704 (**35**), CCDC 2251707 (**38**), and CCDC 2098093 (**56**). These data can be obtained free of charge from The Cambridge Crystallographic Data Center via www.ccdc.cam.ac.uk/data_request/cif. Source data are provided with this paper.

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

## Acknowledgements

We gratefully acknowledge funding from the National Natural Science Foundation of China (grant no. 22371237, 22071209, 22071206), the National Youth Talent Support Program, and the Natural Science Foundation of Fujian Province of China (grant no. 2017J06006).

## Author contributions

L.G. conceived and designed the project, Q.-Y.L., S.C. and T.H. conducted the experiments, Q.-Y.L., F.Y. and L.G. analyzed and interpreted the experimental data, Z.Y. designed and performed the DFT calculations. L.G. and Y.-M.L. prepared the manuscript, Q.-Y.L. and S.C. prepared the Supplementary Information.

## Competing interests

The authors declare no competing interests.
