## [Peer Review File · Nature Communications]

REVIEWER COMMENTS

Reviewer #1 (Remarks to the Author):

This work deals with the development of a sp²-sp³ bond-forming reaction by utilizing FeCl₃ as dual catalyst. The authors suggest that the catalyst participates in the functionalization of C(sp³)-H bonds by generating chlorine radicals that are responsible for a hydrogen-atom transfer event. The corresponding Fe(II) intermediate is later on oxidized by oxygen to recover back the propagating Fe(III) catalyst. In addition, the latter oxidized the carbon-centered radical, thus setting the basis for enabling a Friedel-Craft type reaction.

This work has undoubtedly some strengths. Among them, the employment of inexpensive iron catalyst, the utilization of air as oxidant, and a broad scope of aromatic/heteroaromatic substrates and benzylic precursors. In addition, the authors took the responsibility to look at the reaction behind the scenes with some mechanistic experiments and synthetic applications. That being set, and taking into consideration the inherent interest in designing dual catalytic endeavors for the activation of both sp² and sp³ C-H bonds, I believe that the manuscript would be well suited for Nature Communications. However, a number of non-negligible issues should be brought forward:

(1) As judged by the optimization included into the manuscript, other metal catalysts could be employed for similar purposes. According to a recent paper by Rovis (JACS 2021, 143, 2729), one might argue that the utilization of CuCl in combination with LiCl might be beneficial.

(2) Table 1E: the conditions are not shown in the main text

(3) The optimization should include an entry with O₂ as oxidant

(4) According to the literature data, an acidic media is beneficial for enabling HAT processes. However, one might wonder why the authors haven't looked at the role of the acid in detail.

(5) Scope: (a) no examples are included with electron-withdrawing groups. Even though the Friedel-Craft might not be favored, readers of the journal would appreciate information about these examples into the text; (b) the authors only include indoles or pyrroles as nitrogen-containing heterocycles. Furans, thiophenes or derivatives should be tested as well; (c) As for the formation of 69 and 70, a rationalization behind the selectivity should be given. It is certainly interesting that a secondary carbocation is favored over a tertiary one; (d) as for 69, the authors describe a selectivity of 3:1; however, such a selectivity is calculated by ¹H-NMR and the aromatic signals match perfectly with 10 protons. Please, clarify; (e) The utilization of simple toluene as benzylic precursor should also be considered; (f) The scope should include allylic derivatives not conjugated to an arene, for

example trans-1-heptene. It would be highly instructive to know whether the major compound is linear or branched; (g) regarding the utilization of cycloalkanes, it is somewhat interesting that simple cycloalkanes require harsh conditions. It would be important to include the utilization of simple aliphatic precursors such as n-hexane, and the selectivity found in these cases. In addition to this, the authors need to test the utilization of more complex alkanes such as isopentane in order to know whether the HAT from chlorine radicals might exert site-selectivity principles

(6) The kinetic isotope effect experiments do not include the indicated time. This is important given that the KIE is calculated on a kinetic basis.

(7) Synthetic application: the reaction en route to 107 was conducted in a solvent-free conditions. It would be highly instructive to include the result under the standard reaction conditions.

(8) Supporting information: The following compounds are not pure, and therefore need to be re-purified (if so, the yields need to be adjusted): 6, 9, 11, 13, 14, 16, 18, 19, 20, 21, 22, 23, 24, 26, 27, 28, 31, 32, 33, 36, 37, 38, 39, 40, 41, 42, 48, 49, 53, 55, 57, 61, 62, 64, 65, 66, 67, 68, 76, 88, 89, 90, 94, 99, 100, 101, 102, 105.

Reviewer #2 (Remarks to the Author):

This publication describes a simplified strategy that utilizes multiple functionalities of iron(III) halides under visible light conditions for the direct and selective coupling of low-reactive C(sp²)-H and C(sp³)-H bonds. This protocol produces various C(sp²)-C(sp³) coupling products in good yields and with high chemo- and site-selectivity. Mechanism study is not particularly revealing, but there is nothing obviously missing either. This manuscript would be suitable for publication in Nature Communications after some corrections:

1) The 56 X-ray diagram is not clear, please replace it. (Table 2)

2) Please supplement the compound (1zh-1zm) characterization data (HRMS/IR/M.P.), Also, please replace many unclear ¹H NMR and ¹³C NMR spectrograms (substrate 9, 11, 14, 24, 25, 28 etc.)

3) Why did the nucleophilic reagent (OAc⁻) not attack the benzyl position of the aryl group in the cation capture experiment? (Fig. 3a, right. and Fig. 3e).

4) Gram scale reaction needs to be attempted.

Reviewer #3 (Remarks to the Author):

The HAT processes of sp^3C-H bonds mediated by halide radicals have been extensively studied very recently, taking advantage of the LMCT phenomenon of iron/copper halides upon visible light irradiation. The resulting alkyl radicals were either caught up to form versatile C-X chemical bonds directly, or converted by the radical-polar crossover pathway into the anions/cations before forming the new bonds. The radical-polar crossover reactions of alkyl radicals have also been frequently demonstrated by the latest research works, such as the iron and visible light mediated Friedel-Crafts reactions (Chem 2023, 9, 1610-1621). All together, it is less likely that the manuscript has enough novelty to publish in Nature Communications.

That will be better if the author could provide the rationale of the products 85 and 87, and the corresponding radical and cation species involved.

As Fig.3d shown, the light absorption of $FeCl_3$ fade gradually from 358 to 500 nm. However, the light resource with the max wavelength at 410 nm was superior to those at 395, 455 nm.

The yields were generally low. What kind of side products were generated?

Point-by-Point Response to Referees

Responses to the comments of reviewer 1.

This work deals with the development of a sp^2 - sp^3 bond-forming reaction by utilizing $FeCl_3$ as dual catalyst. The authors suggest that the catalyst participates in the functionalization of $C(sp^3)$ -H bonds by generating chlorine radicals that are responsible for a hydrogen-atom transfer event. The corresponding $Fe(II)$ intermediate is later on oxidized by oxygen to recover back the propagating $Fe(III)$ catalyst. In addition, the latter oxidized the carbon-centered radical, thus setting the basis for enabling a Friedel-Craft type reaction. This work has undoubtedly some strengths. Among them, the employment of inexpensive iron catalyst, the utilization of air as oxidant, and a broad scope of aromatic/heteroaromatic substrates and benzylic precursors. In addition, the authors took the responsibility to look at the reaction behind the scenes with some mechanistic experiments and synthetic applications. That being set, and taking into consideration the inherent interest in designing dual catalytic endeavors for the activation of both sp^2 and sp^3 C-H bonds, I believe that the manuscript would be well suited for Nature Communications. However, a number of non-negligible issues should be brought forward.

Our response:

We express our sincere appreciation to reviewer 1 for the positive evaluation and highly valuable comments, which have served as a strong motivation for us to enhance the manuscript by addressing details that were previously overlooked. In the revised manuscript, we have conducted additional studies on the reaction conditions and expanded the scope of substrate application. These efforts serve to unequivocally demonstrate the universality of our study and further elucidate its strengths.

(1) As judged by the optimization included into the manuscript, other metal catalysts could be employed for similar purposes. According to a recent paper by Rovis (*JACS* **2021**, 143, 2729), one might argue that the utilization of $CuCl_2$ in combination with $LiCl$ might be beneficial.

Our response:

Thank you for bringing up this highly important point. In response to your suggestion, we performed the reaction $1a+2a\rightarrow 3$ under the identical conditions as described in the literature (*JACS* **2021**, 143, 2729). Regrettably, no product was observed under these circumstances. We hypothesize that the absence of product formation might be attributed to two factors: insufficient oxidation of benzyl radicals into the corresponding carbon cations by copper(II) species, and the potential incompatibility of these conditions with subsequent steps involved, such as the Friedel-Crafts process. We have incorporated this important finding into entry 14 in **Supplementary Table 1** of the revised ESI (page S13).

(2) Table 1E: the conditions are not shown in the main text.

Our response:

Thank you sincerely for bringing this to our attention. We deeply apologize for the oversight. To address this issue, we have made the necessary corrections to the labeling of reaction conditions (a–d) in Table 1 in the revised version (page 7).

(3) The optimization should include an entry with O_2 as oxidant.

Our response:

Thank you for your suggestion. We performed the reaction using O_2 as an oxidant, which resulted in a reduced yield of 50%. This finding has been included as entry 22 of **Fig. 2** in the revision manuscript (page 5).

(4) According to the literature data, an acidic media is beneficial for enabling HAT processes. However, one might wonder why the authors haven't looked at the role of the acid in detail.

Our response:

Thank you very much for your helpful comments, which have motivated us to delve deeper into the impact of acidic additives on the reaction. Our findings revealed that the inclusion of protonic acids, such as HCl , AcOH , or CF_3COOH , generally resulted in significantly lower yields. Furthermore, the addition of Lewis acids like ZnCl_2 or $\text{Cu}(\text{OTf})_2$ led to slightly lower yields compared to the standard reaction conditions. As a result, incorporating either protonic or Lewis acids did not improve the reaction outcome, potentially due to compatibility issues with other steps involved in addition to HAT processes. We have duly included these findings as entries 8–13 in **Supplementary Table 1** of the revised ESI (page S13).

Entry	Acidic additive	Yield (%)
1	none	76%

2	HCl in H ₂ O (1.0 M)	trace
3	HCl in Et ₂ O (1.0 M)	33%
4	AcOH	trace
5	CF ₃ COOH	37%
6	ZnCl ₂	73%
7	Cu(OTf) ₂	60%

(5) About scope

(5a) no examples are included with electron-withdrawing groups. Even though the Friedel-Craft might not be favored, readers of the journal would appreciate information about these examples into the text;

Our response:

Thank you for bringing up this crucial point. Following your suggestion, we thoroughly investigated the reactivity of benzene derivatives incorporating electron-withdrawing groups like fluorine, chloride, acetyl, ester, and sulfonyl. Regrettably, our findings revealed that none of these substrates exhibited any reactivity under the standard conditions. We have duly included these results, along with other relevant data, as **3.3 Unsuccessful Substrates** in the revised ESI (page S72).

(5b) the authors only include indoles or pyrroles as nitrogen-containing heterocycles. Furans, thiophenes or derivatives should be tested as well;

Our response:

As suggested, we proceeded to investigate the reaction between furan and thiophene under standard conditions. Regrettably, neither of them exhibited any progress. Our assumption is that furan and thiophene were unable to tolerate the presence of the iron salt during light irradiation. We have included these results, along with other relevant data, as **3.3 Unsuccessful Substrates** in the revised ESI (page S72).

(5c) As for the formation of **69** and **70**, a rationalization behind the selectivity should be given. It is certainly interesting that a secondary carbocation is favored over a tertiary one;

Our response:

Thank you for your suggestion. We have added a statement ‘In these instances, the C(sp³)-H arylation was observed to predominantly take place at secondary C-H bonds, rather than primary or tertiary ones. This selectivity can be attributed to a delicate balance between steric hindrance and electronic preferences,^[ref] as well as the influence of iron salts in other steps of the process.’ in page 8 of the revised manuscript. As emphasized in a notable reference (*Nature* **2016**, 533, 230–234), the functionalization of aliphatic C-H bonds exhibits a preference for primary C-H bonds in terms of steric factors, while tertiary C-H bonds are somewhat more electronically favored. However, it is worth noting that reactions can also take place at secondary C-H bonds due to the delicate balance between steric hindrance, electronic preferences, and catalyst control. This significant literature has been appropriately cited as ref. 41.

Notably, due to the inclusion of additional examples, we have made an adjustment in the numbering of the original compounds. As a result, the previous compounds numbered **69** and **70** have been updated to number **70** and **71**, respectively, to ensure accurate sequencing.

(5d) as for **69**, the authors describe a selectivity of 3:1; however, such a selectivity is calculated by ¹H-NMR and the aromatic signals match perfectly with 10 protons. Please, clarify;

Our response:

Thank you for bringing this issue to our attention. We have promptly rectified it and made the necessary corrections in the ESI (page S166).

(5e) The utilization of simple toluene as benzylic precursor should also be considered;

Our response:

Thank you for this valuable suggestion. We conducted an investigation on the reaction between toluene and phenol, and we were able to obtain the desired products with a yield of 50%. This result has been appropriately incorporated into **Table 2** (product **67**).

(5f) The scope should include allylic derivatives not conjugated to an arene, for example trans-1-heptene. It would be highly instructive to know whether the major compound is linear or branched;

Our response:

Thank you for your suggestion. We proceeded with an investigation into the reaction involving trans-1-heptene under standard conditions. Regrettably, we did not observe any significant progress with this substrate. Our working hypothesis is that allylic derivatives that are not conjugated to an arene display limited reactivity under standard conditions. In our revised ESI, we have included these results along with other pertinent data in section **3.3 Unsuccessful Substrates** (page S72).

(5g) regarding the utilization of cycloalkanes, it is somewhat interesting that simple cycloalkanes require harsh conditions. It would be important to include the utilization of simple aliphatic precursors such as *n*-hexane, and the selectivity found in these cases. In addition to this, the authors need to test the utilization of more complex alkanes such as isopentane in order to know whether the HAT from chlorine radicals might exert site-selectivity principles.

Our response:

Thank you for emphasizing this crucial point. Taking your suggestion into account, we conducted an extensive investigation using a range of simple aliphatic precursors, such as *n*-hexane, as well as more complex alkanes. Regrettably, our efforts only yielded minimal product quantities. We propose that competitive reactions, particularly self-coupling, may be more favorable under these circumstances. To provide a comprehensive overview, we have incorporated these findings and other pertinent data into section **3.3 Unsuccessful Substrates** in the revised ESI (page S72).

(6) The kinetic isotope effect experiments do not include the indicated time. This is important given that the KIE is calculated on a kinetic basis.

Our response:

Thank you for raising this concern. We have now incorporated the reaction time (10 h) for the Kinetic Isotope Effect (KIE) experiments in Figure 3g of the updated manuscript on page 12. Additionally, we have

included this information in the revised ESI on page S83.

(7) Synthetic application: the reaction en route to **107** was conducted in a solvent-free conditions. It would be highly instructive to include the result under the standard reaction conditions.

Our response:

As per your suggestion, we proceeded with the reaction using the standard conditions and observed a decrease in yield compared to solvent-free conditions. The obtained result has been integrated into manuscript into **Fig. 4b** (page 14).

A description has been incorporated in page S72-73 of the revised ESI as shown below:

A solvent-free reaction: A Schlenk tube (10 mL) was charged with 4-methoxyindole (**1zv**, 839.8 mg, 5.7 mmol), cyclohexane (**2zh**, 20 mL) and FeBr₃ (591.0 mg, 0.11 mmol). The mixture was degassed *via* three freeze-pump-thaw cycles, then filled with dry air. The Schlenk tube was positioned approximately 5 cm away from a 40 W LED lamp ($\lambda_{\text{max}} = 370$ nm). After being stirred at 25 °C for 120 h, the reaction mixture was concentrated to dryness. Purification using silica gel column chromatography (elution with PE:DCM = 2:1) gave the pure product **108** as a white solid (444.1 mg, 1.938 mmol, 34% yield).

A reaction in dichloromethane: A Schlenk tube (10 mL) was charged with 4-methoxyindole (**1zv**, 839.8 mg, 5.7 mmol), cyclohexane (**2zh**, 4.82 g, 57 mmol) and FeBr₃ (591.0 mg, 0.11 mmol). The mixture was degassed *via* three freeze-pump-thaw cycles, then filled with dry air. The Schlenk tube was positioned approximately 5 cm away from a 40 W LED lamp ($\lambda_{\text{max}} = 370$ nm). After being stirred at 25 °C for 120 h, the reaction mixture was concentrated to dryness. Purification using silica gel column chromatography (elution with PE:DCM = 2:1) gave the pure product **108** as a white solid (156.6 mg, 0.684 mmol, 12% yield).

(8) Supporting information: The following compounds are not pure, and therefore need to be re-purified (if so, the yields need to be adjusted): 6, 9, 11, 13, 14, 16, 18, 19, 20, 21, 22, 23, 24, 26, 27, 28, 31, 32, 33, 36, 37, 38, 39, 40, 41, 42, 48, 49, 53, 55, 57, 61, 62, 64, 65, 66, 67, 68, 76, 88, 89, 90, 94, 99, 100, 101, 102, 105.

Our response:

Thank you for bringing attention to this crucial point. We have meticulously purified the mentioned products and subsequently adjusted the yields and spectra accordingly. The updated results can be found in both the revised manuscript and ESI.

Responses to the comments of reviewer 2.

This publication describes a simplified strategy that utilizes multiple functionalities of iron(III) halides under visible light conditions for the direct and selective coupling of low-reactive C(sp²)-H and C(sp³)-H bonds. This protocol produces various C(sp²)-C(sp³) coupling products in good yields and with high chemo- and site-selectivity. Mechanism study is not particularly revealing, but there is nothing obviously missing either. This manuscript would be suitable for publication in Nature Communications after some corrections:

Our response:

We genuinely appreciate the positive evaluation and valuable comments provided by reviewer 2, which have motivated us to enhance this study further. In the revised manuscript, we have undertaken additional

research on the reaction mechanism, effectively illustrating the universality of this study and elucidating its strengths.

1) The **56** X-ray diagram is not clear, please replace it. (Table 2)

Our response:

Thank you for raising this concern. We have addressed it by replacing the previous content with a clear X-ray diagram, which is now included in **Table 2** of the revised manuscript.

2) Please supplement the compound (**1zh-1zm**) characterization data (HRMS/IR/M.P.), Also, please replace many unclean ^1H NMR and ^{13}C NMR spectrograms (substrate **9**, **11**, **14**, **24**, **25**, **28** etc.)

Our response:

We genuinely appreciate your meticulous examination of our characterization data. As per your suggestion, we have included the characterization data of **1zh-1zm** in the revised ESI (page S4–8). Moreover, we have conducted a re-purification of unclean substrates such as **9**, **11**, **14**, **24**, **25**, **28**, and some products, resulting in updated yields and spectra.

3) Why did the nucleophilic reagent (OAc^-) not attack the benzyl position of the aryl group in the cation capture experiment? (Fig. 3a, right. and Fig. 3e).

Our response:

Thank you for bringing up this crucial matter, which motivated us to delve into the intricacies of these experiments. According to a relevant literature reference (*Angew. Chem. Int. Ed.* **2019**, *58*, 8577), we proposed a plausible mechanism for the formation of product **88**. Initially, cyclopropylbenzene (**87**) undergoes a single electron transfer with the chlorine radical generated in situ, resulting in the formation of a radical cation (**Int-A**) and a chloride anion. Subsequently, the intermediate **Int-A** is subjected to nucleophilic attack by either the chloride anion or the acetate ion, leading to the formation of a ring-opening radical species (**Int-B** or **Int-C**). The final step involves H-abstraction from the substrates or other reaction components, ultimately leading to the formation of either product **88** or a chloride analog. The latter can then undergo an $\text{S}_{\text{N}}2$ reaction with NaOAc to yield **88**. Although this mechanism initiates with a SET rather than an HAT process as proposed in the photochemical $\text{C}(\text{sp}^2)\text{-H}$ alkylation, it can indirectly suggest the involvement of chlorine radicals.

The proposed reaction mechanism has been incorporated to the revised ESI (page S78).

A related ref. (from **87** to **Int-B**): *Angew. Chem. Int. Ed.* **2019**, *58*, 8577

4) Gram scale reaction needs to be attempted.

Our response:

Thank you for your suggestion. We have successfully conducted a gram-scale reaction of 4-methoxy-1*H*-indole + cyclohexane → **108**. This result has been incorporated into **Fig. 4b** for reference (page 14).

Responses to the comments of reviewer 3.

The HAT processes of sp^3 C–H bonds mediated by halide radicals have been extensively studied very recently, taking advantage of the LMCT phenomenon of iron/copper halides upon visible light irradiation. The resulting alkyl radicals were either caught up to form versatile C–X chemical bonds directly, or converted by the radical-polar crossover pathway into the anions/cations before forming the new bonds. The radical-polar crossover reactions of alkyl radicals have also been frequently demonstrated by the latest research works, such as the iron and visible light mediated Friedel-Crafts reactions (*Chem* **2023**, 9, 1610-1621). All together, it is less likely that the manuscript has enough novelty to publish in Nature Communications.

Our response:

We deeply appreciate the highly valuable comments provided by Reviewer 3, which have inspired us to enhance the manuscript and emphasize the robustness of this study. While a recently reported visible light-mediated Friedel-Crafts reaction exists, **our research endeavors to tackle a persistent challenge regarding the direct and selective cross-coupling of a strong C(sp^2)-H bond and a robust C(sp^3)-H bond, specifically targeting the coupling of C(sp^2)-H bonds in benzenes with C(sp^3)-H bonds in cycloalkanes.** This task is notably difficult due to the formidable bond strength and the challenge of achieving desirable chemo- and site-selectivity. Only very limited studies have reported methods to address this issue, usually relying on the utilization of metal-modified zeolite, montmorillonite materials, harsh conditions, and/or specific substrates.

Our study represents the first successful demonstration of a general and mild C(sp^2)-H/C(sp^3)-H coupling of benzenes with aliphatic hydrocarbons like cycloalkanes, utilizing catalytic amounts of iron halides as multifunctional activators and air as a green oxidant. This step-, cost-efficient approach facilitates access to highly valuable building blocks from abundant starting materials, and provides a sustainable alternative to conventional methods in organic synthesis.

In response to the feedback from Reviewer 3, we have made necessary revisions in the updated manuscript to further elucidate this point. A statement **'Under mild conditions, the reaction between a strong C(sp^2)-H bond and a robust C(sp^3)-H bond has been achieved, affording a broad range of cross-coupling products with high yields and commendable chemo-, site-selectivity.'** has been incorporated in the abstract.

1) That will be better if the author could provide the rationale of the products **85** and **87**, and the corresponding radical and cation species involved.

Our response:

Thank you for bringing up this important issue, which motivated us to delve into the intricacies of these experiments. Due to the inclusion of additional examples, we have made an adjustment in the numbering of the original compounds. As a result, the previous compounds numbered **85** and **87** have been updated to number **86** and **88**, respectively, to ensure accurate sequencing.

Regarding the formation of product **86**, its mechanism appears to be intricate and lacks clarity. We hypothesize that it involves a hydrogen atom abstraction by a chlorine radical, leading to the generation of a benzyl radical, followed by a radical ring-opening reaction and a subsequent series of oxidation/reduction processes.

Regarding the formation of product **88**, we propose a plausible mechanism according to a relevant literature reference (*Angew. Chem. Int. Ed.* **2019**, *58*, 8577). Initially, cyclopropylbenzene (**87**) undergoes a single electron transfer with the chlorine radical generated in situ, resulting in the formation of a radical cation (**Int-A**) and a chloride anion. Subsequently, the intermediate **Int-A** is subjected to nucleophilic attack by either the chloride anion or the acetate ion, leading to the formation of a ring-opening radical species (**Int-B** or **Int-C**). The final step involves H-abstraction from the substrates or other reaction components, ultimately leading to the formation of either product **88** or a chloride analog. The latter can then undergo an S_N2 reaction with NaOAc to yield **88**. Although this mechanism initiates with a SET rather than an HAT process as proposed in the photochemical C(sp²)-H alkylation, it can indirectly suggest the involvement of chlorine radicals.

The proposed reaction mechanism has been incorporated to the revised ESI (page S78).

A related ref. (from **87** to **Int-B**): *Angew. Chem. Int. Ed.* **2019**, *58*, 8577

2) As Fig.3d shown, the light absorption of FeCl₃ fade gradually from 358 to 500 nm. However, the light resource with the max wavelength at 410 nm was superior to those at 395, 455 nm.

Our response:

Thank you for bringing this to our attention. The UV-Vis absorption spectrum clearly shows that the iron salt exhibits significant absorption at wavelengths greater than 400 nm. We conducted experiments with various light sources emitting different wavelengths as shown below, and the results indicated that the optimal light source is a 410 nm LED.

Several related studies (*Angew. Chem., Int. Ed.* **2020**, *59*, 23603–23608; *J. Am. Chem. Soc.* **2020**, *142*, 6216–6226; *Nat. Commun.* **2021**, *12*, 2377) have highlighted the importance of considering the maximum absorption peak when selecting an appropriate light source. However, it is worth noting that using a light source with the maximum absorption wavelength of a photoactive substance may not always yield maximum benefits. This could be attributed to the fact that an appropriate rate of radical formation is more advantageous for reactions compared to a fast rate. The optimized light source conditions have been included in **Supplementary Table 1** of the revised supplementary information (page S13).

Entry	Light source	Yield (%)
1	$\lambda_{\text{max}} = 370 \text{ nm}$	37
2	$\lambda_{\text{max}} = 395 \text{ nm}$	56
3	$\lambda_{\text{max}} = 410 \text{ nm}$	76
4	$\lambda_{\text{max}} = 425 \text{ nm}$	60
5	$\lambda_{\text{max}} = 440 \text{ nm}$	33
6	$\lambda_{\text{max}} = 455 \text{ nm}$	24

3) The yields were generally low. What kind of side products were generated?

Our response:

Thank you for your highly valuable comment, which has encouraged us to conduct a more detailed analysis of the reaction. We thoroughly examined the side-products resulting from the reactions of **1a+2a**→**3** and **1p+2a**→**18**, and identified small amounts of chloro-substituted products and self-coupling products (**91**, **92**, **111** and **112**). This important result has been incorporated as **3.5 Identification of Side Products** (page S73). A statement 'In the reaction of **1a+2a**→**3**, trace amounts of chloro-substituted products (**91**, **111**) and a self-coupling product (**92**) were detected and identified. Similarly, in the reaction of **1p+2a**→**18**, analogous side-products (**91**, **92**, **112**) were observed.' as well as the characterization data of the new compounds **111** and **112**, have been added in this section.

Other changes:

1. Dr. Yu-Mei Lin has made substantial contributions during the revision of this manuscript, including conducting thorough analysis and addressing all comments raised by the reviewers. Additionally, she has provided guidance to students in experiment design and implementation. Consequently, she is being designated as a co-corresponding author for this study in the revised version.
2. Page numbers have been incorporated into the revised manuscript.
3. We have corrected the spelling of the word "*ortho*" and highlighted it in yellow in the revised manuscript (on pages 2 and 6).
4. Due to the inclusion of ref. 41, ref. 43, and ref. 44, the reference numbers in the manuscript have been adjusted accordingly.

REVIEWERS' COMMENTS

Reviewer #1 (Remarks to the Author):

The authors have submitted a new revised manuscript. From the cover letter, it is evident that the authors have looked carefully into all issues posed by the reviewers. That being set, I believe that the paper is now suited for Nature Communications

Reviewer #2 (Remarks to the Author):

The author has addressed the concerns of the reviewers and the article can be published without further modifications

Reviewer #3 (Remarks to the Author):

My concerns have been addressed in the revised manuscript.